# How Do Health Teams Perceive International Migrant Users of Primary Care? [note 1]

**DOI:** 10.3390/ijerph19169940

**Published:** 2022-08-12

**Authors:** Consuelo Cruz-Riveros, Alfonso Urzúa, Gustavo Macaya-Aguirre, Báltica Cabieses

**Affiliations:** 1Escuela de Psicología, Universidad Católica del Norte, Antofagasta 1270709, Chile; 2Escuela de Enfermería, Facultad de Salud, Universidad Santo Tomás, Concepción 3460000, Chile; 3Facultad de Ciencias Sociales, Universidad Alberto Hurtado, Santiago 8320000, Chile; 4Facultad de Medicina Clínica Alemana, Universidad del Desarrollo, Santiago 7610658, Chile

**Keywords:** health personnel, transients and migrants, primary health care

## Abstract

The following study aims to describe the perception of international migrants who use the primary level of care by health personnel and to explore how this perception can affect the care process in the commune of Antofagasta, north of Chile. *Methodology:* The methodology was qualitative using a phenomenological descriptive design, through which the discourses of the health personnel of three primary health care centers (*n* = 14) were explored. *Results:* The participants presented two thematic categories. The first one related to experiences during delivery of care to migrants and included perceptions and beliefs, factors associated with using services, and reasons for consultation. The second category related to stereotypes towards migrants, including prejudices, beliefs about their lives and reasons for migration. *Conclusions:* The therapeutic relationship with migrants in primary care depends on the health care personnel’s acceptability of this population, hence, allowing the delivery of care based on respect for their culture and rights.

## 1. Introduction

Migration is defined by the World Health Organization (WHO) as the movement of people to a different territory, regardless of its size or composition [1]. This movement can be explained by multiple causes such as educational, political or socioeconomical (poverty, inequality, and job insecurity), natural disasters, health-related issues and family needs. Moreover, deciding to migrate will sometimes be associated with a conjunction of several factors that trigger migration [1,2,3].

The context in which migration takes place is generally associated with complicated historical processes from the country of origin and sometimes the receiving country [4,5]. However, the migrant world population has been increasing since the 1990s [2,3]. There is a growing interest in its study as a social, political and economic factor of human and social development, and raising awareness on human rights is a cornerstone dimension to consider among people on the move [4,5]. Statistically, since 2020 the population of the Americas reached 7.2% of the global population, while in South America, it reached 2.5% of the total population [4,5,6,7]. In Chile, migrants represent 8.6% of the population; the countries with the greatest presence in Chile are Venezuela (30.7%), Peru (16.3%), Haiti (12.5%), Colombia (11.4%) and Bolivia (8.5%) [6,7]. Regarding territorial distribution, the three regions with the highest percentage of foreign residents are: Metropolitan, with 63.1% of foreigners residing in the country; Antofagasta, with 7.1%; and Valparaíso, with 6.4%. Antofagasta is one of the regions with a greater weight on the total regional population, with 13.6% [6,7,8,9,10]. For its part, the main migratory flow in the country is intra-regional and most movements in past years have come from sectors of the Colombian coast and Venezuela, characterized by a large majority of Afro-descendants [8].

The interest in promoting respect for human rights in past decades has reached the world’s capacity to generate international human rights pacts that most countries in Latin America have supported. This in turn has promoted the development of policies that aim at guaranteeing respect for every migrant person and the delivery of basic social protection including health care in receiving countries. Such policies and strategies aim to articulate actions for the protection of people on the move that might be exposed to multiple human rights violations during the migration process or during settlement in the host country [2,3]. The most reported types of human rights violations among migrant communities in the region are discrimination, poor working conditions, overcrowded housing, lack of basic sanitation, violence, contaminated environments and health inequities [8,9,10,11,12,13].

The exposure to multiple social and economic risk factors by international migrants in receiving countries can generate short-, medium-, or long-term deterioration in their health status, which is why the WHO includes migration as a social determinant of health [14]. Among the strategic indicators disseminated by the WHO for this population is the right to health, which establishes a set of minimum and universal services required for the fulfillment of the maintenance or improvement of their health [15]. Such services must include availability of resources and action for gender inequities in health, address distinctions based on the stage of the life course and take into account their particular culture. It also considers the time and geographical distance for effective accessibility to care, and discrimination based on preconceived prejudices or stereotypes that can be observed in the healthcare system, including those reproduced by health teams [16,17,18].

Some of the practices recommended by experts and health authorities in Chile include language adaptation, consideration of intercultural approaches, and addressing the social determinants of health of the international migrant population [9,19]. The purpose of these standards of care is to respond to the person’s own needs and improve human-rights-oriented and culturally pertinent care at the different levels of the health system [15,19,20,21,22,23,24]. The implementation of these standards is challenging and continuously developed at different areas and levels of the healthcare system. In the meantime, the literature reports factors that can act as facilitators or barriers to the use of health services by international migrants. Facilitators that come from the healthcare system are, for example, efforts to reduce gaps in access to care, including updates for available services and actions to increase migrant registration in the healthcare system [21,22,23,24]. On the other hand, there are a number of barriers to healthcare, such as a lack of resources and limited implementation of regulations and programs that promote sensitive care. These results, in some cases, in practices based on personal criteria in which health personnel beliefs, prejudices (positive or negative), and in some cases, xenophobia emerge [21,22,23,24]. One example of negative beliefs towards migration reported in the existing literature is related to the idea that the undocumented migrants are “illegal” and do not deserve access to healthcare [25]. This could reflect a lack of knowledge and critical reflection on the part of health personnel about who the undocumented migrant is and his/her specific risks and needs [24]. Another example is discrimination to some migrants who come from some specific countries of origin that are considered of “lower caste”. This is due to stereotypes historically forged in popular imaginaries about the different migratory groups in the country, like people coming from Peru, Haiti and Bolivia. In addition, there is literature suggesting the undocumented migrant users are considered more demanding for healthcare than other communities [15,21,26,27].

In social psychology, prejudice is an attitude that has three components: (i) the cognitive component associated with beliefs and the positive or negative assessment of the “stereotypes” of an outgroup, (ii) the component related to feelings and emotions towards the outgroup and (iii) the component related to behavior that can be positive or negative towards the outgroup [28,29]. Stereotypes are produced by generalized evaluations of simplified mental images or ideas about characteristics and attributes of the people that make up a group, based on cultural beliefs, and even generating prejudices based on stereotypes [28]. Another of its characteristics is the generalized assessment of people who are defined as a member of such a prejudiced group or community, guiding their attitudes and behaviors as if they all were the same [28]. Prejudice can also manifest itself overtly or subtly [29]. The first refers to traditional prejudice, characterized by perceived threats to resources, resulting in open rejection and contact with the outgroup [29]. The second is based on the defense of traditional values and where the perception of the outgroup is one of disrespect and of receiving undeserved benefits [29]. In short, people who experience negative feelings toward international migrants may present a negative stereotype about them and possibly discriminatory acts [28,29].

Based on the foregoing, and considering that the behavior towards international migrants during health care may be based on the perception, beliefs or prejudices that health providers have about this population, the following question arises: What is the perception by health personnel of international migrants who use primary care in Chile? Could their perceptions about international migrants affect the healthcare delivery process? Therefore, the objective of this research was to describe the perception by health personnel of international migrants who use the primary level of care in Chile and to explore how this perception might affect the process of care. We considered the commune of Antofagasta, located in the north of Chile and with the second highest concentration of international migrants in the country, as the setting for this study. This borough was especially affected during the COVID-19 pandemic as thousands of migrants entered illegally by foot though the Atacama Desert in the border between Chile, Peru and Bolivia, and ended in transient camps in the city of Antofagasta and other cities close by [10].

## 2. Method

### 2.1. Design

The study’s methodology was qualitative, as it aims at exploring and unveiling the phenomenon of interest in its natural environment, trying to make sense of such phenomena through the perspectives of the people that are a part of it. This methodology allows the construction of the knowledge and understanding of people’s behavior, assuming that reality is multiple, dynamic and changing [30,31,32]. The type of investigation is descriptive phenomenological, through which we investigate the thematic contents of the discourses of health personnel from five primary health care centers in the commune of Antofagasta in the north of Chile.

### 2.2. Participants

The sampling strategy was intentional using the snowballing technique and continued until saturation of qualitative information was reached. A total of 14 people participated voluntarily (8 in individual interviews and 6 in a single focus group). The participants belonged to five different public primary health care centers in the commune of Antofagasta, Chile. Data collection was carried out from April to August 2021 in person at the centers. The inclusion selection criteria considered were: (i) over 18 years of age and (ii) have provided health services to international migrants in the past.

### 2.3. Data Collection

Data was collected through multiple individual interviews and one focus group (Table 1). Semi-structured scripts were developed for each of these qualitative methodologies in order to provide in-depth information about the perception of health personnel concerning providing care to international migrants at the primary level. Conducting in-depth interviews reduced potential distractions or the desirability of responses produced by pressure or threats from other participants, as can occur with focus groups [31]. It also secures that every participant has sufficient time to express all of his/her ideas and experiences, something that can be challenging in focus groups [30,32]. In this study, the focus group was considered a complementary strategy of data collection to individual interviews, allowing the research team to review the relevance and significance of categories that emerged in the interviews and exploring potential new ones during the group conversation. Both the interviews and the focus group were carried out in the workplace, guaranteeing COVID-19 preventive safety measures and a calm environment that ensured confidentiality of participants, yet allowing for a face-to-face interaction.

The participants were selected using the snowball sampling technique, starting with health workers invited to participate in the study at the primary centers during recruitment. Those who accepted to participate became the seed participant who then recommended other health workers to participate. We chose this sampling technique due to the difficulty of accessing health personnel; they were not easy to contact directly. However, they felt more confident and comfortable when a colleague spoke to them for the first time. The participants of the interviews and the focus group were different, to ensure a wide range of points of view. Participation was voluntary; before obtaining their consent, participants were informed about the research and their rights as participants in this research.

Semi-structured scripts for interviews and the focus group included flexible questions based on study objectives and were intended to delve into the perception that healthcare teams expressed about “being an international migrant”. The scripts were revised by academic experts before piloting, after which some minor adjustments were made, and data collection proceeded. Some of the questions used were: How has the experience of providing care to international migrants in primary care been? Are there any specific healthcare protocols for migrants that have been established in this health center? What actions have you implemented to promote or improve care towards migrants? Are there any stereotypes about migrants among health workers in this center? What are they? How could these stereotypes, in your opinion, affect healthcare delivery to this population? The interviews lasted for 40 to 60 min and were audio recorded, transcribed verbatim, and then thematically analyzed based on main pre-defined categories from the scripts, yet allowing for emerging categories to be captured and analyzed.

### 2.4. Rigor

Scientific rigor in qualitative research can be performed by several techniques. One is triangulation, a method in which various sources of narrative information are contrasted. In this study, we included the triangulation of types of participants. The triangulation of perspectives included those provided by health personnel, intercultural referents, and authorities from the primary care centers [33]. A second rigor technique, audit trial, in our study was developed though a second researcher who took notes during data collection and followed up the path of decisions used during the investigation, reaching similar conclusions in terms of data saturation [34]. A third element of scientific rigor is reflectivity, which is performed through an exhaustive revision of steps and decisions made by the research team, especially during the analysis and interpretation phase.

### 2.5. Ethical Considerations

This research had the approval of the ethics committee of Universidad del Desarrollo (protocol code 2019-094 and approval date 22 November 2019) and was governed by the principles of voluntariness, autonomy and confidentiality of participants as well as protection of participants and data. It included the signing of an informed consent before data collection.

### 2.6. Data Analysis

Once the transcripts of the interviews were fully available in a Word document, the thematic analysis was carried out as it has been developed in similar previous studies [32,34]. The thematic analysis included first the repeated reading of transcriptions to identify main categories and related themes. The broad categories were coded from the matrix through the construction of analysis categories assigned by the researchers, assigning them one or more codes related to their meaning and intention. After that general descriptive phase, a process called “decanting” was carried out, following the open coding, then the axial coding and to finish later with the selective coding [35,36]. After analysis of interviews was carried out, we conducted the same analysis process for the focus group and then both techniques were integrated into a single analysis matrix. QTY. 1. New NVivo version 1.5 software for MAC (Antofagasta, Chile) was used to code the interviews.

## 3. Results

The study participants’ perceptions related to providing care to international migrants in primary level included two main categories: (i) beliefs about the migration process, factors associated with the use of health services and reasons for consultation; and (ii) stereotypes based on personal beliefs and, in some cases shared by more than one health team member, concerning international migrants. The presence of prejudices about international migrants and migration itself could represent positive or negative ideas that could intervene in the healthcare delivery process regarding the disposition towards the resolution of the health needs of international migrant patients.

(1) Experiences of health teams related to international migrants during health care delivery: beliefs about the migration process, factors associated with the use of health services and reasons for consultation.

The participants reported having had previous experiences of healthcare delivery towards international migrants in the primary center. One of the most frequent aspects of such experience was related to their social integration process and their living conditions, which in their perspective proved to be relevant to address their particular health needs. Thus, according to primary care health workers, the causes of migration might go beyond mere labor and individual issues and could be oriented towards a better quality of life for themselves and their families. Some of the triggering factors for migration mentioned by the study participants were the search for better education, housing, health, family reunification and access to health. According to the health workers, migrants also report as relevant causes for the decision to migrate those concerning sociopolitical problems faced by their countries of origin.

*“… They came from Peru because they suffered from meningitis there and did not receive medical support, they did not have as much access… Their migration was mainly for access to health so that their son would receive better health…”*.(EP1)

*“….. they are looking for a better quality of life, education, health and housing, they also come for family unification…. Venezuelans attribute it to the political-social issue…. the Bolivian or Peruvian population has never told me something like that, only for education, health or housing...”*.(EP5)

*“...In general, migrants, mostly women, heads of households and people like in the case of Colombia, Peru and Bolivia, relatively young people, 30 years old, right?... Venezuela is an overwhelmed case, so to speak...”*.(EP2)

*“... Most of the cases of patients who are foreigners are single mothers, there are very few who are in a couple, because sometimes the husband is there taking care of the children, she comes, she does not know that she is pregnant...”*.(EP3)

Based on previous experiences providing primary care to international migrants at their centers, study participants were able to identify factors that positively and/or negatively influenced the quality of care they provided and the continuity of care. They highlighted the lack of knowledge, on both sides, about each other’s worldviews and understandings of health-related processes. Health care workers for example perceived poor knowledge and training regarding intercultural health and knowing cultural aspects necessary for the adaptation of interventions to the needs of international migrant communities. Also, according to study participants, migrant populations were unaware of the Chilean health system, its characteristics and availability of services. This lack of knowledge was often associated with a lack of understanding of the information, which could sometimes be explained by their educational level or differences with the health system from the country of origin (for example, the lack of primary care in their home country). Another factor that participants identified was language, as the ability to handle other languages and intercultural adaptations of existing health-related information (pamphlets, posters, etc.). A fourth factor, sometimes derived from those mentioned above, was that beliefs and ignorance triggered staff mistreatment towards international migrants. Finally, study participants described structural and historical weaknesses of the primary centers such as long waiting lists, financial constraints and staff shortage. All of these were also affecting the opportunity of access to care among both migrants and locals.

*“...health professionals, in general, are not very aware of the issue, we must understand that there is a significant cultural shock, they have different customs, even different idioms, which interfere with our communication, that is why there are to be super empathic with them…”*.(EP3)

*“…A patient who was Bolivian, from the highlands, I assumed she was Aymara, and she came as with her papers… the person who had done her papers did not explain anything to her, and there is a cultural issue there… she did not speak Spanish well”*.(EP1)

*“...there are many who are informed and who know how to access health care, but there are many who are not informed...”*.(EP6)

(2) Stereotypes towards migrants, often based on personal beliefs from healthcare teams.

According to study participants, the deficit of resources and the poor understanding of the problems that have triggered the massive arrival of international migrants to Antofagasta in the past two years of pandemic are crucial to comprehend the reality of these communities and their health-related needs. In this regard, study participants visualize a high demand of health care from international migrants in the city in general and in primary care. Some health workers even connect such “relatively higher” health demand among migrants as a possible cause for a reduction in the rate of delivery of care for the local population.

*“…Peru or Bolivia tends to be very inward, but clearly the violence is much greater…. to control boys and girls”*.(EGF2)

*“Psychomotor development, I have had a lot of children lately… I have had a lot of children with a lot of delayed psychomotor development, and it is because they do not come, and they are not interested in the child having these tests or those controls up to date”*.(EGF4)

*“….It depends a lot on the nationality of the foreigner, I have had more of the vision of the Bolivian foreigner, he is a little more negligent with his health”…“They wait until the end, they come like this when the thing is not to prevent, but rather Now to cure, I think that in the dental issue it is like this the point to come to remove the tooth”*.(EGF5)

*“…. It is a population that one as a Chilean does not fully understand…. That is why if I could define them, I would say not rare, because it is not the specific word, but special, another way of understanding health which is what brings us here. I would also perhaps define them as very demanding; they tend to be poly-consultants and a little impatient”*.(EGF6)

Regarding stereotypes, the participants recognize the existence of negative ideas against migrants in the general society, including xenophobia and machismo. Some arguments that could explain the existence of stereotypes, according to study participants, are related to generalized views of foreigners based on nationality and their use of services. For example, health workers reported the perception that some types of migrants, based on country of origin, are less prone to self-care including basic hygiene and negligence of their offspring. These ideas are deeply embedded in their conceptions related to some migrant communities such as those coming from Bolivia or Peru. Health workers perceived that these types of ideas have an effect on the ways health care is delivered to them at the primary level.

*“…on a professional level… I remember that recently an official told me mmm… hey, all the migrants come to get pregnant…” “… the other day I heard a colleague say <<… super abusive with their children>>, and I have Bolivian patients who are a seven like mom”*.(EP1)

*“…Venezuelans still have a lot of tools because they have a lot of studies and like they are very easy, docile in terms of their personality, that’s why it’s like it’s easy for them to find work… the Bolivian and Peruvian population emm… obviously their personality a lot Sometimes it’s like more submissive…”*.(EP2)

*“…To the people of Bolivia, Peru, you have to explain more slowly, you have to ask them more times… “do you understand me? “yes”, “no?”. “…With Xenophobia, with machismo, especially from employers, because being a woman is already difficult for this society, imagine being a woman and apart from being a migrant*.(EP3)

*“… They are very grateful, I have never been touched by a migrant person who has not been satisfied or who has given them the same, they are never, never grateful”*.(EP4)

*“...A Bolivian person costs us a little more because sometimes they don’t understand very well what one asks, they are also much shyer, they don’t consult as much, and Peruvians are also a bit like that...”*.(EP6)

*“... It is often the case that professionals believe they have the moral property of judging the patient for having passed through a border crossing...”. “...Bolivians, because they have a slightly humbler culture, are not very good at dealing with some things”*.(EP7)

## 4. Discussion

The study explores the perception of international migrants who use public primary care in the city of Antofagasta in the north of Chile by health personnel, and how such perceptions could affect their interactions with this population. According to the WHO, international migration is considered a social determinant of health, and the risks experienced during the move and at arrival might deteriorate the health status of international migrants [37]. Traditionally, healthcare systems all over the world require considering the context in which the individual is inserted and their personal life trajectory, worldview and health-related beliefs, thereby investigating protective and risk factors that might affect health outcomes in a given context [37,38,39,40,41,42]. Among the risk factors related to the poor health of international migrants reported are: the relevance of socioeconomic and material living conditions (overcrowding, basic sanitation), working conditions, lack of social support, and others [41]. According to Hernando, cited by Aninat and Vergara [41], some accumulated deficiencies in the migrant communities that affect their health are the product of factors such as ignorance of their rights or poor governmental aid. In addition, they include those associated with their life experiences at their countries of origin that can lead to poor health over their life course, such as limited vaccination schemes, poor social protection, disasters and economic crises. Therefore, it is possible to appreciate that the migration status by itself is not a risk factor, but rather the associated conditions are the ones that can trigger health problems [27]. International migrants’ particular experiences and needs could be considered as a relevant aspect of the improvement of healthcare to protect the health of all.

This study reported that health personnel’s perceptions about providing care to international migrants might present facilitating factors and barriers. According to our findings, many of these facilitators and barriers could be interpreted from the lens of ethical clinical practice and human rights. From the guidelines provided by the WHO, it is essential to consider the right to health approach in health care for international migrants, where the dimension of availability and accessibility are relevant, but it is the acceptability of culturally sensitive quality care that needs further attention and development. Such dimension of universal coverage establishes the need to respect differences in culture, language, gender, migration status and other aspects as an essential articulating axis between therapeutic relationships in primary care [15].

Barriers to effective healthcare among international migrants have been widely studied, involving individual aspects such as limited information about the healthcare system in the receiving country, fear of deportation, language, lack of health insurance and educational level. Our study and other similar ones in the international literature suggest the relevance of paying attention to unethical practices in healthcare delivery, such as dehumanized treatment, communication problems, violation of the right to care, discrimination and health services without cultural adaptation to specific diverse communities [24,26,41,42,43]. According to Liberona, the relationship between the health personnel and international migrant users could be intervened by these factors, reproducing scenarios of power asymmetry between both actors [21]. Health personnel have the power to grant or decline the possibility of health care to migrants (i.e., a violation of the right to health) and migrants could be underpowered due to poor understanding of the health system and how to navigate it to find solutions to their unique health needs [21]. From the healthcare system and the health teams’ perspectives, it is necessary to recognize the current shortcomings for the development of the necessary skills to be open to actions that allow diverse cultural practices during care delivery [42]. Current practices use a homogenizing criterion that ignores the differences between international migrants, not considering their diversity in nationality, language, ethnicity, migration process, country of origin, migration status, family context, and the adaptation, integration or marginalization they have faced while residing in Chile [24].

In all, when thinking of power relations in healthcare settings, the literature suggests some relevant elements that seem to emerge in the therapeutic relationship between the healthcare teams and international migrants. Such elements might act as facilitators or barriers to access, as described by various authors worldwide [37,38,39]. The dynamics produced in these therapeutic contexts can be influenced by experiences and knowledge built throughout life by each of the parts involved, in which the systems of beliefs, traditions, language and cultural codes act as mediators or blockers in the perceptions of how the healthcare process is being developed [38]. According to Roter, relations must be based on the acknowledgment from all actors of the principle of mutual respect and horizontal communication [39]. A therapeutic relationship that ignores these principles often risks the emergence of negative outcomes like fear, discrimination and limited continuity of care among migrants [38].

The findings from this study indicate that staff willingness to provide care to international migrants might be mediated by personal knowledge, beliefs or voluntariness in daily practice. A factor that acts as a barrier to care would be the lack of existence of standardized practices or that these are not known to all staff in primary care and in the healthcare system in general. For Avaria, this creates a barrier to access the health system, which can lead to a poor recognition of migrants as subjects of right by health providers [42].

The dimension of stereotypes found in our study tended to represent more negative than positive aspects of the relationship between healthcare workers and migrant patients. We found that some generalizations associated with nationality were linked by health workers to the migrants’ ability to follow instructions, self-care practices and others. These findings are consistent with previous studies on negative beliefs and prejudiced attitudes related to many migrants’ perception of job loss, opportunities and low quality of health care due to unsensitive or poorly culturally pertinent care provided to them [42,43,44]. According to Pino’s study, each health team member’s vision can be biased when providing care in the therapeutic relationship and can affect the quality of care [45].

The perception of health personnel regarding the international migrant population can affect their relationship with the health system more broadly. For instance, undergraduate students in training can learn through modeling about negative stereotypes and prejudices against migrants that they could reproduce later in their professional life. In fact, some studies show that negative attitudes are associated with the year of study: the lower the course, the better the attitude towards migrants [43]. The international literature has indicated that such attitudes are embedded in beliefs that connect migrants with delinquency, the collapse of the health system, excessive alcohol consumption and increased health spending [46]. It is relevant to know the perception of health personnel about the international migrant population and how it can positively and negatively influence health practices. Therefore, it is imperative to promote strategies and actions that involve both professionals and students in training for acquiring knowledge and skills that delve into ethical aspects and interculturality in order to reduce negative attitudes and prejudices that act as barriers to health care for migrants.

This study has a number of limitations. First, the scope is restricted to the city of Antofagasta in the northern area of Chile. Second, it does not include migrants form all relevant countries of origin, yet over 70% of them are from countries within the region. In future research, it would be pertinent to expand the sample to the country’s different regions, include the different types of health personnel (auxiliary, administrative and professional) and different levels of care (secondary, tertiary).

## 5. Conclusions

To achieve progress in health care, training and awareness on ethical principles of healthcare delivery in contexts of social and cultural diversity and practical intercultural competencies for healthcare teams is fundamental. For this, it is necessary to recognize structural and historical shortcomings in most health systems when it comes to ethical, human-rights-oriented and culturally sensitive care for international migrants. These structural limitations trigger the failure to respond in a timely and adequate manner to the needs of international migrant populations. As a result, most health workers tend to homogenize migrant communities as a single, unified culture. When this occurs, pre-conceived ideas and attitudes such as stereotypes and prejudices act as a powerful barrier to effective healthcare. Despite the health team’s lack of financial support or knowledge, they reported the willingness and commitment to provide more effective and culturally pertinent care according to local possibilities and realities.

## Figures and Tables

**Table 1 ijerph-19-09940-t001:** Description of study participants.

Health Centers	Code	Profession	Gender	Age	Years of Experience
1	EP3	Matron	Female	28	Greater than 1 year *
1	EP4	Administrative	Female	56	30
1	EP7	Social worker	Female	28	2
2	EP1	Psychologist	Female	29	4
2	EP6	Nurse	Female	39	6
3	EP2	Psychologist	Female	36	5
4	EP5	Social worker	Female	31	6
4	EGF1	Nursing Technician	Female	33	9
4	EGF2	Physician	Female	25	2 months
4	EGF3	Matron	Male	28	2
4	EGF4	Nursing	Female	36	36
4	EGF5	Psychology	Female	28	6 months
4	EGF6	Dentist	Male	31	3 months
5	EP8	Technical reference	Female	Over 18 *	Greater than 1 year *

* They do not indicate the exact time.

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
