# Peer review of "How Do Health Teams Perceive International Migrant Users of Primary Care?â€"

_ijerph, 2022, doi:10.3390/ijerph19169940_

Round 1
Reviewer 1 Report
My suggestion was to group personnel by their center instead of listing of types of personnel and the number of centers that had the personnel category.I had also thought if you could match the center with the immigrants they that they met with and based their feedback on you would have a greater insight about their perceptions and what may have infuenced their perceptions. Your current table provides numbers and types of personnel.
When you first read your reasearch question,it sounds as if ths is going to be a quantitative study especially using terms such as analyze and sampling. You want to do the study to better understand the perceptions...
Author Response
We appreciate the review of the article, and its recommendations are included in the manuscript.
"Please see the attachment."

Reviewer 2 Report
In the revised manuscript, the authors adequately addressed my comments posed on the previous version of their manuscript. I reckon the revised manuscript now can deliver clearer scientific messages to readers of this journal.
Author Response
We appreciate the review of the article, and its recommendations are included in the manuscript.
"Please see the attachment."

This manuscript is a resubmission of an earlier submission. The following is a list of the peer review reports and author responses from that submission.
Round 1
Reviewer 1 Report
Thank you for the opportunity to review this study of healthcare professionals’ attitudes to migrant care users in Chile.
There are some potentially interesting data in the paper; however, I’m unsure how much the paper contributes beyond identifying negative attitudes towards migrants, attributed to others, and reported by a small sample of healthcare workers. Is there anything the authors can say about how these attitudes impacted on care?
The presentation of the paper is the main issue. At present, the arguments are hard to follow, and the findings feel underdeveloped. The paper also needs a thorough proof-read, there are some very long sentences that are particularly hard to follow, random capitalisation and random full-stops (periods).
I have suggestions for improvements.
The methods section requires more detail. What is meant by a ‘phenomenological-descriptive interpretive study’? Is there a reference for this?
Why was snowball sampling used intentionally? How were participants identified? Why were some interviewed in a focus group?
More detail is needed on data analysis; e.g. was double-coding undertaken to ensure rigour?
The findings feel under-developed. The quotes aren’t well-integrated into the findings, making it hard to follow the ‘story’. Potentially rich and interesting phenomena are skimmed over; e.g. pg 5, lines 194-20: a range of factors are listed but not developed.
A couple of sections of the findings would be better placed in the introduction (pg 5, lines 193-201) or the discussion (pg 6, lines 264-274).
I found the discussion very hard to follow.
There is no consideration given the study’s limitations. For example, the risk of bias arising from snowball sampling, the risk of socially desirable reporting.
I’m not convinced of the extent to which the findings shed light of how healthcare is affected by the attitudes expressed by participants.
Author Response
We appreciate the review of the article, and its recommendations are included in the manuscript. Please see the attachment.
- Writing: Extensive editing of English language and style required
Response: Thank very much for your review. The revision of the language and style in English was incorporated. Please see the attachment.
Study Design:
- The methods section requires more detail. What is meant by a ‘phenomenological-descriptive interpretive study’? Is there a reference for this?
Response: References were included, and this area was improved.
The study's methodology is qualitative, as it presents the study of things in their natural environment, trying to make sense of the phenomena being interpreted from people's perspectives. This methodology allows the construction of knowledge and understanding of people's behavior, assuming that reality is dynamic and changing [32,36]. The type of investigation is descriptive – phenomenological, through which the discourses of health personnel from three Primary Health Care centers in the commune of Antofagasta, Chile, are explored.
- Why was snowball sampling used intentionally? How were participants identified? Why were some interviewed in a focus group?
Response: The details of the selection of participants are included. The methods of data collection (focus group and interview) are deepened, in addition to the sample selection.
Participants
The type of sampling strategy was intentional by snowball until the saturation of the speech. A total of 14 people participated voluntarily (8 in individual interviews and 6 in a focus group). The participants belonged to five different primary health care centers in the commune of Antofagasta, Chile; the information collection was carried out from April to August 2021. The inclusion criteria considered were: Over 18 years of age and have provided health services to international migrants.
Data collection
The data was collected through individual interviews and focus groups (table 1). The collection with these two techniques was structured to learn in-depth the perception of health personnel about international migrants who use the first level of care and to explore how this perception can affect the care process. Conducting in-depth interviews reduces potential distractions or the desirability of responses produced by pressure or threats from other participants, as can occur with focus groups [31]. Another benefit of the in-depth interviews is the minimum bias compared to the focus groups. Among the relevant characteristics is the time allocated in the interviews for each participant where equality is sought versus the focus groups that time participation may be disproportionate in some cases [32,33]. Both the interviews and the focus group were carried out in the workplace, guaranteeing health guarantees and a calm environment that ensured the total confidentiality and anonymity of the participants, allowing face-to-face interaction with the participant.
The participants were selected using the snowball sampling technique; the availability made the selection at a given time of the seed participant that will be recommended to other participants. The selection of participants through this sampling is also due to the difficulty of accessing health personnel; they were not easy to contact directly. However, they felt more confident and comfortable when a colleague spoke to them for the first time. The participants in the interviews and focus groups were different to ensure a wide range of points of view. Participation was voluntary; before obtaining their consent, they were informed about the research and their rights as participants in this research.
- More detail is needed on data analysis, e.g. was double-coding undertaken to ensure rigour?
Response: The requested elements are included in the manuscripts.
Rigor
Rigor is guaranteed by triangulation, where various sources guide the selection of study participants. The perspectives presented include health personnel, intercultural referents, and authorities. The method used was focus groups and individual interviews that presented semi-structured questions (34). A second aspect is an audit; to ensure the study's rigor, another researcher followed up the path of decisions used, reaching similar conclusions (35). A third element is a reflection, contemplated through permanent review, especially during the analysis and interpretation phase of the information. Expert researchers on the subject reviewed the generation of the question script.
Ethical considerations
This research has the approval of the ethics committee of the Universidad del Desarrollo (protocol code 2019-094 and approval date November 22, 2019) and was governed by the principles of voluntariness, confidentiality, and anonymity of the participants, which is reflected through the signing of informed consent by all the research participants.
Data analysis
Once the transcripts of the interviews were made, the thematic analysis was carried out, which is consistent with previous studies [33,35]. The phases that were carried out begin with the transcription and then the transcripts' repeated reading to identify possible themes. The broad themes were coded from the matrix through the construction of analysis categories assigned by the researchers, based on the participants' narratives. Each interview sheet marks the words and phrases, assigning them one or more codes related to their meaning and intention. Thus, a process called "decanting" is carried out, following the open coding, then the axial coding to finish later with the selective coding [36]. NVivo version 1.5 software for MAC was used to code the interviews.
- The findings feel under-developed. The quotes aren’t well-integrated into the findings, making it hard to follow the ‘story’. Potentially rich and interesting phenomena are skimmed over; e.g. pg 5, lines 194-20: a range of factors are listed but not developed.
- A couple of sections of the findings would be better placed in the introduction (pg 5, lines 193-201) or the discussion (pg 6, lines 264-274).
Response: The first paragraph is redone to guide the understanding of the results better.
The perception of health personnel during care experiences includes perceptions and beliefs about the migration process, factors associated with the use of services, and reasons for consultation. In addition to the above, the staff presents stereotypes based on personal beliefs and, in some cases, shared by more than one health team member. The presence of prejudices about the international migrant and migration can present positive or opposing ideas, which can intervene in the care process regarding the disposition towards the resolution of health needs.
- I found the discussion very hard to follow.
Response: Most of the discussion is restructured in the manuscript
The study reported the perception of health personnel about migrants of other nationalities, who use the primary level of care, under the assumption that this could affect the care process. According to the WHO, international migration is considered a social determinant of health, constituting an actual or potential risk for the deterioration of the health status of the international migrant population [39]. Traditionally, health care requires considering the general context in which the individual is inserted, thereby investigating protective and risk factors, which may be the cause of current living conditions or may also be related to the migration process [39,40, 42-44]. Among the risk factors are the conditions in which they live (overcrowding, basic sanitation), working conditions, and state of physical and mental health [39]. According to Hernando, cited by Aninat and Vergara [43], some accumulated deficiencies of the migrant are the product of factors such as ignorance of their rights or government aid. In addition, they include those associated with their origins that can lead to multidimensional poverty. Therefore, it is possible to appreciate that the migratory condition by itself is not a risk factor, but rather the associated conditions are the ones that can trigger health problems [23]. International migrants consider in care experiences the need to delve into this population's life history and health needs, thus contributing to global knowledge of the causes of migration, customs, and strategies to implement in health practices.
The study reported that the perception of health personnel about international migrants might present facilitating factors or barriers in the care process. Among the factors reported with the most significant reference are those that involve ethical aspects of health practices. From the guidelines provided by the WHO, it is essential to consider the right to health approach in health care for the international migrant population, where the dimension of acceptability establishes respect for culture, language, stage of the life cycle, and gender as an articulating axis between therapeutic relationships [11].
The barriers have been extensively studied, involving individual aspects such as ignorance, fear of deportation, lack of health insurance and educational level. From the health system, there are ethical elements such as dehumanized treatment of the provider, lack of right to care, discrimination and health benefits without cultural adaptation [20,22,43]. According to Liberona [17], the relationship between health personnel and international migrant users could act as part of these factors, which would mediate the exercise of power asymmetrically between both actors. The result by the health personnel, there would be the power to grant the possibility of health (violation or enjoyment of the right to health) and the migrant would present a lack of knowledge regarding the achievement of the resolution on the health need presented [17]. From the health system and the health team, it is necessary to recognize the current shortcomings for the development of the necessary skills to open to actions that allow diverse cultural practices [38]. Current practices use a homogenizing criterion that ignores the differences between international migrants, not taking into account their nationality, language, ethnicity, migration process and adaptation they have faced [20].
The speeches of the participants give an account of the knowledge of regulations and documents for the health care of international migrants. According to this, the perception of the staff or the willingness to care presented by each team member can be mediated by personal knowledge, beliefs, or voluntariness in daily practice. A factor that acts as a barrier to care would be the lack of standardized practices known to all staff. For Avaria, this creates a barrier to access to the health system, which can lead health providers not to recognize these people as subjects of rights [38].
The dimension of stereotypes, present positive and negative elements directly related, mostly with generalizations associated with nationality linking it with understanding ability, educational level, performance care otres, and self-care. Among the harmful elements is the loss of opportunities for the native population in health centers. The general negative perception is attributed to other health team members and can be found in contrast to the positive connotations. The findings are consistent with previous studies on negative beliefs and prejudiced attitudes related to many migrants' perception of job loss. opportunities, and low quality of health care due to the care provided to the migrant population [44-46]. According to Pino's study, each health team member's vision can be biased when providing care [47].
The perception of health personnel regarding the international migrant population can affect care processes in establishments. Among the factors that undergraduate students in training can address are stereotypes and prejudices. Studies show that negative attitudes are associated with the year of study, where the lower course, the better attitude [45]. The themes that have triggered such individual and collective beliefs are associated with migration and: delinquency, the collapse of the health network, excessive alcohol consumption, and increased health spending [48]. It is relevant to know the perception of health personnel about the international migrant population and how this appreciation can positively and negatively influence health practices. Therefore, it is imperative to develop measures involving both professionals and students in training for acquiring knowledge and skills that delve into ethical aspects and include interculturality to reduce negative attitudes and prejudices that act as barriers in health care.
- There is no consideration given the study’s limitations. For example, the risk of bias arising from snowball sampling, the risk of socially desirable reporting.
Response: The request is included in the manuscript.
The study's limitations are as follows: the scope is restricted, mainly given the geographical limitation of the study and the countries of origin of the vast majority of international migrants from northern Chile are mainly from South America. In future research, it is pertinent to expand the sample to the country's different regions and include the different levels of health personnel (auxiliary, administrative, and professional). In addition to the above, the design limitations do not allow the results to be generalized. Regarding researcher bias, in this type of research it is about minimizing; that is why the declaration of the predetermined procedures and the use of information collection methods.
- I’m not convinced of the extent to which the findings shed light of how healthcare is affected by the attitudes expressed by participants.
Response: The discussion and conclusion are worked on to achieve an adequate approach to the results.
CONCLUSION
To achieve progress in health care, the training and awareness of the health team members become a fundamental aspect for the development of acceptability that seeks health practices based on respect for others as a fundamental element. For this, it is necessary to recognize individual and collective shortcomings as health teams and as a health system.
The preceding inevitably triggers a failure to respond in a timely and adequate manner to the needs of the international migrant. As a result, we have general practices that tend to homogenize some aspects of the care process, guarding against standards and programs that do not achieve adequate implementation according to the reality of the health center. When these shortcomings occur, we find elements such as stereotypes or prejudices that act as barrier factors in care. One of the implemented strategies is the intercultural facilitator, recognized as relevant support in primary health centers. Despite the health team's lack of money and knowledge, there is a willingness and commitment to providing health care according to local possibilities and realities.

Reviewer 2 Report
Authors:
I appreciate the opportunity to review your article which I found to be very interesting and it provided similar results to articles I have read concerning individuals, especially individuals of color attempting to migrant into the United States using the Southern border. I offer the following comments:
Writing:
· I noticed grammatical errors, which can be rectified by use of an editor. In some cases, there were periods in the wrong places, run on sentences, and incorrect word usage.
Study Design:
· Use of qualitative research was appropriate for the study.
· Your research study question was somewhat awkward and could be written in a clarifying manner.
· Were interview and focus group participants provided consent forms to participate?
· Your study participants) health professionals) were listed in order of years of experience. A chart/table lumping those of similar years of experience together and the those that they interviewed would provide more insight into the responses provided by the professional team. You may also want to consider putting the care team together if there was more than one. The way they are currently listed, the reader cannot determine if this as one care team or if this was more than one care team or if different clinics were used. Then match the participants to the care team. Quite often a dominating member of the care team can influence the perceptions, attitudes and/or biases of the entire team.
Article Content
· I did not read any limitations to the study, e.g., researcher bias or the acknowledgement that this is merely a snapshot of what was felt during this given time period.
· I am not sure you can draw the conclusions you did during the relatively short time period used to collect the data. I am a little concerned that you (the authors) had some pre-conceived thoughts of what you wanted the article to show.
Author Response
We appreciate the review of the article, and its recommendations are included in the manuscript. Please see the attachment
- Writing: I noticed grammatical errors, which can be rectified by use of an editor. In some cases, there were periods in the wrong places, run on sentences, and incorrect word usage.
Response: Thank very much for your review. The revision of the language and style in English was incorporated.
x
Study Design:
- Your research study question was somewhat awkward and could be written in a clarifying manner.
Response: Work on writing research questions.
Based on the foregoing, and considering that the behavior towards international migrants during health care may be based on the perception, beliefs or prejudices that health providers have from the health system, the following question arises: What is the perception of health personnel about international migrants who use the first level of care? Could perception affect the attention process? Therefore, the objective of this research is to analyze the perception of health personnel about international migrants who use the primary level. of attention and explore how this perception can affect the process of attention in the commune of Antofagasta, Chile.
- Were interview and focus group participants provided consent forms to participate?
Response: They are included in the manuscript area of ​​ethical considerations the details of the signed consent of participants in the research.
This research has the approval of the ethics committee of the Universidad del Desarrollo (protocol code 2019-094 and approval date November 22, 2019) and was governed by the principles of voluntariness, confidentiality, and anonymity of the participants, which is reflected through the signing of informed consent by all the research participants.
- Your study participants) health professionals) were listed in order of years of experience. A chart/table lumping those of similar years of experience together and the those that they interviewed would provide more insight into the responses provided by the professional team. You may also want to consider putting the care team together if there was more than one. The way they are currently listed, the reader cannot determine if this as one care team or if this was more than one care team or if different clinics were used. Then match the participants to the care team. Quite often a dominating member of the care team can influence the perceptions, attitudes and/or biases of the entire team.
Response: They are included in the manuscript, The methodology in the participating items indicates the relevance to 5 different centers of the interviewees. This information is included in greater detail in Table 1. Please see the attachment
The type of sampling strategy was intentional by snowball until the saturation of the speech. A total of 14 people participated voluntarily (8 in individual interviews and 6 in a focus group). The participants belonged to five different primary health care centers in the commune of Antofagasta, Chile; the information collection was carried out from April to August 2021. The inclusion criteria considered were: Over 18 years of age and have provided health services to international migrants.
Table 1.
Code |
Health centers |
Profession |
Gender |
Age |
Years of experience |
EP1 |
2 |
Psychologist |
Female |
29 |
4 |
EP2 |
3 |
Psychologist |
Female |
36 |
5 |
EP3 |
1 |
Matron |
Female |
28 |
Greater than 1 year * |
EP4 |
1 |
Administrative |
Female |
56 |
30 |
EP5 |
4 |
Social worker |
Female |
31 |
6 |
EP6 |
2 |
Nurse |
Female |
39 |
6 |
EP7 |
1 |
Social worker |
Female |
28 |
2 |
EP8 |
5 |
Technical reference |
Female |
Over 18* |
Greater than 1 year * |
EGF1 |
4 |
Nursing Technician |
Female |
33 |
9 |
EGF2 |
4 |
Physician |
Female |
25 |
2 months |
EGF3 |
4 |
Matron |
Male |
28 |
2 |
EGF4 |
4 |
Nursing |
Female |
36 |
36 |
EGF5 |
4 |
Psychology |
Female |
28 |
6 months |
EGF6 |
4 |
Dentist |
Male |
31 |
3 months |
Article Content
- I did not read any limitations to the study, e.g., researcher bias or the acknowledgement that this is merely a snapshot of what was felt during this given time period.
Response: Incorporated into the manuscript.
The study's limitations are as follows: the scope is restricted, mainly given the geographical limitation of the study and the countries of origin of the vast majority of international migrants from northern Chile are mainly from South America. In future research, it is pertinent to expand the sample to the country's different regions and include the different levels of health personnel (auxiliary, administrative, and professional). In addition to the above, the design limitations do not allow the results to be generalized. Regarding researcher bias, in this type of research it is about minimizing; that is why the declaration of the predetermined procedures and the use of information collection methods.
2. I am not sure you can draw the conclusions you did during the relatively short time period used to collect the data. I am a little concerned that you (the authors) had some pre-conceived thoughts of what you wanted the article to show.
Response: Incorporated into the manuscript. Please see the attachment
To achieve progress in health care, the training and awareness of the health team members become a fundamental aspect for the development of acceptability that seeks health practices based on respect for others as a fundamental element. For this, it is necessary to recognize individual and collective shortcomings as health teams and as a health system.
The preceding inevitably triggers a failure to respond in a timely and adequate manner to the needs of the international migrant. As a result, we have general practices that tend to homogenize some aspects of the care process, guarding against standards and programs that do not achieve adequate implementation according to the reality of the health center. When these shortcomings occur, we find elements such as stereotypes or prejudices that act as barrier factors in care. One of the implemented strategies is the intercultural facilitator, recognized as relevant support in primary health centers. Despite the health team's lack of money and knowledge, there is a willingness and commitment to providing health care according to local possibilities and realities.

Reviewer 3 Report
This is a study to address the perception of health professional about primary health care use of migrants. The findings of this study could contribute to global health settings.
I would recommend authors to describe specific question items asked in their semi-structured interviews.
Author Response
1. Este es un estudio para abordar la percepción de los profesionales de la salud sobre el uso de la atención primaria de salud de los migrantes. Los hallazgos de este estudio podrían contribuir a la configuración de la salud global.
Recomendaría a los autores que describieran las preguntas específicas formuladas en sus entrevistas semiestructuradas.
Respuesta: Muchas gracias por su revisión. Hemos incorporado todas sus sugerencias en el manuscrito. Consulte el archivo adjunto.
2. Se requieren cambios moderados en inglés.
Respuesta: Muchas gracias por su revisión. Se incorporó la revisión del lenguaje y estilo en inglés. Consulte el archivo adjunto.

Round 2
Reviewer 1 Report
The authors have made good efforts to improve the manuscript in some areas. However, there is still some proofreading and correction of typos required.
Little effort has been made to address my original comments:
- The findings feel under-developed. The quotes aren’t well-integrated into the findings, making it hard to follow the ‘story’. Potentially rich and interesting phenomena are skimmed over; e.g. pg 5, lines 194-20: a range of factors are listed but not developed.
- A couple of sections of the findings would be better placed in the introduction (pg 5, lines 193-201) or the discussion (pg 6, lines 264-274).
I am still of the opinion that the findings section requires major revisions.
Author Response
We thank you once again for reviewing the article. The recommendations made were integrated into the manuscript
Writing:
- However, there is still some proofreading and correction of typos required.
Response: The correction of typos was incorporated.
Study Design:
- Little effort has been made to address my original comments:
- The findings feel under-developed. The quotes aren’t well-integrated into the findings, making it hard to follow the ‘story’. Potentially rich and interesting phenomena are skimmed over; e.g. pg 5, lines 194-20: a range of factors are listed but not developed.
Response: The request is included in the manuscript.
Reports from public entities such as the Municipal Corporation for Social Development of Antofagasta have indicated that of the total PHC users in the Antofagasta commune, 13.05% are international migrants within the groups with the largest population are Colombians, Bolivians, Peruvians, and Venezuelans, where 72.14% are women [30]. From the historical background, since the 1990s, Chile has presented an increase in migration, characterized by female migration, in addition to the indigenous population from the north and, in general, from neighboring countries and other Latin American countries [30].
- A couple of sections of the findings would be better placed in the introduction (pg 5, lines 193-201) or the discussion (pg 6, lines 264-274).
Response: The request is included in the manuscript. Please see the attachment.
The literature has delved into issues that seem to manifest themselves globally in the relationship and communication in a therapeutic context between the health team and migrants and may act as barriers to access, as described by various authors worldwide [39-41]. The dynamics produced in these therapeutic spaces can be influenced by experiences and knowledge built throughout life, where the system of beliefs, traditions, language and codes act as mediators in the perceptions of the actors [40]. According to Roter, relations must be based on the acknowledgment of both actors under the principle of mutual respect and horizontal communication [41]. Among the consequences associated with factors such as cultural respect, language adaptation, immigration status, fear of mistreatment, and discrimination in the late use of health care services in emergencies by migrants [40].
